# Takotsubo Cardiomyopathy After Cytoreductive Surgery and Hyperthermic Intraperitoneal Chemotherapy for a Recurrent Colon Cancer: A Life-Threatening Complication

**DOI:** 10.3390/diagnostics14212402

**Published:** 2024-10-28

**Authors:** Bogdan Moldovan, Iris-Iuliana Adam, Radu-Mihai Pisică, Vlad Untaru, Doly Stoica, Alexandra Șpac, Irina Modrigan, Mihai Ursu, Liliana Jupoiu, Adina Frâncu, Florentina Pescaru, Amir Hubeishie, Adriana Zolog, Liliana Vecerzan

**Affiliations:** 1General Surgery Department, ‘St. Constantin’ Hospital, 500299 Brasov, Romania; adamirisiuliana@gmail.com (I.-I.A.); radu.pisica@spitalulsfconstantin.ro (R.-M.P.); 2Faculty of Medicine, “Lucian Blaga” University, 550169 Sibiu, Romania; liliana.novac@ulbsibiu.ro; 3Anesthesia and Intensive Care Unit, ‘St. Constantin’ Hospital, 500299 Brasov, Romania; vlad.untaru@spitalulsfconstantin.ro (V.U.); stoicadoly@yahoo.com (D.S.); spac.alexandramaria@gmail.com (A.Ș.); 4Anesthesia and Intensive Care Unit, Sanador Hospital, 011031 Bucharest, Romania; irina.modrigan@gmail.com; 5Cardiology Department, ‘St. Constantin’ Hospital, 500299 Brasov, Romania; mihai.ursu@spitalulsfconstantin.ro (M.U.); lylyana_2004@yahoo.com (L.J.); adina_francu@yahoo.com (A.F.); 6Oncology Department, ‘St. Constantin’ Hospital, 500299 Brasov, Romania; florentina.pescaru@spitalulsfconstantin.ro; 7Oncology Department, Timiş County Emergency Clinical Hospital, 300723 Timișoara, Romania; hubeishie@gmail.com; 8Oncology Department, Oncohelp Medical Center, 300425 Timișoara, Romania; 9Pathology Department, Regina Maria Hospital, 400500 Cluj-Napoca, Romania; adizolog@yahoo.com; 10Oncology Department, ‘Dr. Alexandru Augustin’ Military Hospital, 550024 Sibiu, Romania

**Keywords:** Takotsubo cardiomyopathy, stress cardiomyopathy, acute heart failure, Krukenberg tumor, cytoreductive surgery (CRS), hyperthermic intraperitoneal chemotherapy (HIPEC), ventricular fibrillation, perioperative cardiac complications, left ventricular dysfunction

## Abstract

(1) Background: Takotsubo cardiomyopathy, or stress cardiomyopathy, is an acute heart failure condition with transient left ventricular (LV) motion abnormalities but no significant coronary artery obstruction. It mimics acute coronary syndrome (ACS), with symptoms like chest pain, dyspnea, and ECG changes. (2) Case Report: We present the case of a 44-year-old female with relapsed colon cancer and peritoneal carcinomatosis. After undergoing cytoreductive surgery (CRS) and hyperthermic intraperitoneal chemotherapy (HIPEC), she experienced cardiac arrest from ventricular fibrillation 18 h postoperatively. Echocardiography revealed a reduced LV ejection fraction (20%) and apical akinesia, suggesting a Takotsubo Cardiomyopathy. Intensive resuscitation and inotropic support led to gradual recovery. Coronary angiography confirmed no coronary artery obstruction. (3) Discussion: This case highlights TTS as a rare but severe complication following major oncological surgeries, possibly triggered by both physical and emotional stressors. TTS should be considered in the differential diagnosis of perioperative cardiac events in cancer patients. (4) Conclusions: Prompt recognition and management of TTS in the perioperative period are crucial to improving outcomes, especially in high-risk oncological patients undergoing extensive surgeries.

## 1. Background

Takotsubo cardiomyopathy, commonly referred to as stress cardiomyopathy [1], is an acute form of heart failure that manifests as transient left ventricular (LV) wall motion abnormalities, without the presence of significant coronary artery obstruction. First identified in Japan in the early 1990s [1], TTS has drawn attention due to its striking resemblance to acute coronary syndrome (ACS), sharing clinical features such as chest pain, shortness of breath, electrocardiogram (ECG) abnormalities, and elevated cardiac biomarkers [2]. Despite its benign reputation in early research, recent studies have shown that TTS can lead to severe complications, including cardiogenic shock and life-threatening arrhythmias. While TTS typically occurs in postmenopausal women, often following significant emotional or physical stress, its exact pathophysiology remains incompletely understood. Current hypotheses suggest a strong association with excessive sympathetic stimulation, leading to catecholamine-mediated myocardial stunning [1,2]. Additionally, estrogen receptor polymorphisms may play a role in making the cardiovascular system more vulnerable to adrenergic stress, particularly in women. In recent years, an increasing number of TTS cases have been reported in patients undergoing major surgical procedures, including oncological surgeries. Cancer patients, in particular, are exposed to both psychological and physiological stressors, which could increase the risk of TTS in the perioperative period [3]. Despite its growing recognition, TTS remains underdiagnosed in surgical settings [4], where symptoms are mimicking those of ACS or other postoperative complications [5].

## 2. Case Report

This case report details the presentation, diagnosis, and management of a 44-year-old female who experienced a relapse of colonic cancer with peritoneal carcinomatosis in 2021, two years after the initial diagnosis. The patient exhibited specific symptoms indicative of disease recurrence. In 2019, the patient underwent surgical resection and received adjuvant chemotherapy for colonic cancer. Despite an initial period of remission, in 2021 she presented with symptoms such as abdominal distension, persistent abdominal pain, and changes in bowel habits. Imaging studies, including CT scans, confirmed the presence of peritoneal carcinomatosis, indicating a relapse of colonic cancer along with two suspected tumorous formations localized in the ovaries (Figure 1 and Figure 2). Upon relapse, the patient underwent a thorough diagnostic workup, including imaging studies and tumor marker assessments. Elevated levels of carcinoembryonic antigen (CEA) of 5.44 ng/mL and CA-19-9 of 58 U/mL were observed, supporting the diagnosis of colonic cancer recurrence with peritoneal carcinomatosis.

The peritoneal cancer index (PCI) [6] in this case was 14, suggesting a moderate degree of peritoneal surface involvement by the peritoneal carcinomatosis. The peritoneal cancer index (PCI) is a numerical score used during CRS to quantify the extent of disease in different regions of the abdomen and pelvis. With a PCI of 14, there is a moderate volume of disease present in various regions. The significance of this score depends on several factors, including the specific locations of tumor involvement, the type of cancer, and the overall health of the patient [7]. The cardiological evaluation prior to surgery revealed no significant cardiac comorbidities with no ventricular dilatation and an ejection fraction (EF) of 65%. No coronarography was requested, as the patient had no history of past cardiac events (Figure 3).

Due to the peritoneal involvement, the patient underwent CRS in which a 4-quadrant peritonectomy, total hysterectomy with bilateral adnexectomy, omentectomy, splenectomy, and cholecystectomy were performed.

HIPEC using Oxaliplatin 600 mg, a platinum-based chemotherapeutic agent, for 60 min at 42 degrees Celsius was performed. Oxaliplatin is known for its efficacy against colorectal cancer and has been increasingly utilized in HIPEC procedures [8]. The heated intraperitoneal chemotherapy aimed to address both systemic and localized disease, delivering Oxaliplatin directly to the affected peritoneal surfaces. She remained stable throughout the surgery, except for a need for low-dose norepinephrine infusion for hemodynamic stability.

In the postoperative period, approximately 18 h after undergoing surgery for colonic cancer recurrence, the patient experienced an abrupt onset of cardiac arrest because of ventricular fibrillation. Immediate resuscitation measures, including cardiopulmonary resuscitation (CPR), defibrillation, and administration of vasoactive medications, were initiated. The multidisciplinary team, including cardiologists, intensivists, and oncologists, collaborated closely to manage the cardiac complications. Despite resuscitation efforts, the patient faced two instances of heart arrests, after which she developed severe bradycardia, before establishing normal sinus rhythm (Figure 4). The echocardiogram performed after the resuscitation showed an EF of 20% with akinesia of the medium and distal portion of the interventricular anterior septum, the apex, and the medium and distal portion of the lateral ventricular wall with hypokinesis of the ventricular base (an aspect highly suggestive of catecholamine-induced cardiomyopathy or takotsubo cardiomyopathy), moderate mitral regurgitation, acute pulmonary edema and a hypokinetic right ventricle, with no pleural effusion. There was a significant rise in cardiac enzymes.

The decision was made to perform a coronarography, which showed no signs of acute or chronic coronary obstruction (Figure 5 and Figure 6). The patient was put on a continuous infusion of 0.01 micrograms/kg/min of noradrenaline and 1.8 mcg/kg/min of dobutamine with 40 mg of enoxaparin given the next day. She remained intubated for 48 h and the improvement of her hemodynamic status allowed the discontinuation of the inotropic medication the following day. After the steady improvement of the respiratory and cardiac functions, the patient’s clinical status allowed extubation with spontaneous breathing. The patient was started on the standard therapy for cardiac insufficiency, consisting of beta blockers, antiarrhythmic therapy, Angiotensin-converting enzyme (ACE) inhibitors and levosimendan. The following echocardiogram showed slow but steady improvement in the cardiac function. She was discharged from intensive care on day 6 and from the hospital on day 10. The echocardiogram conducted on the day she was discharged from the ICU revealed an EF of 25%, with akinesia of the ventricular apex and medio/apical segments, with no right ventricular dilatation and mild mitral regurgitation. On day 9 after HIPEC, the echocardiogram revealed an EF of 30–35% with hypokinesis of the apex and middle-to-apical segments of the heart. The patient did not complain of angina or dyspnea and tolerated lying in the supine position. She was discharged from the hospital on a combination of beta blockers, antiarrhythmic medication and ACE inhibitors, with a follow-up echocardiogram after one month.

The histopathological findings indicated a widespread metastatic adenocarcinoma of colorectal origin, affecting the peritoneum, omentum, and ovaries, with the ovarian involvement characterized as Krukenberg tumors. These tumors, common in metastatic gastrointestinal cancers, further supported the diagnosis of peritoneal carcinomatosis secondary to colorectal cancer [9]. There was no evidence of primary malignancies in the other organs examined (uterus, spleen), and immunohistochemical staining confirmed the colorectal origin with CDX2 positivity and CK7, PAX8, WT1, and CK20 negativity.

At the 3-year follow-up, the patient remains in complete remission, with no clinical or radiologic evidence of recurrence (Figure 7 and Figure 8).

Periodic imaging, including contrast-enhanced CT scans, and tumor marker evaluations (CEA, CA-19-9) have consistently shown normal levels. This suggests sustained control of both peritoneal carcinomatosis and metastatic disease, demonstrating the long-term efficacy of the aggressive surgical and chemotherapeutic approach despite the severe postoperative cardiac complication [10].

## 3. Discussion

Now, it does seem that the HIPEC played a role in the onset of Takotsubo cardiomyopathy. During cytoreductive surgery, significant fluid loss occurs due to blood loss and visceral exposure over an extended operative period. Hemodynamic alterations may arise from peripheral vasodilation caused by hyperthermia, which accompanies the administration of intraperitoneal chemotherapy. Additionally, the combination of cytoreductive surgery and HIPEC elevates proinflammatory cytokine levels, which leads to further vasodilation and increased vascular permeability, disrupting hemodynamic stability [11]. Oxaliplatin may exert its effects either directly or through its metabolite, oxalate, which chelates calcium and magnesium, potentially interfering with ion channel kinetics. Autonomic nerve dysfunction is another possible aspect of oxaliplatin-induced neurotoxicity, although it typically does not pose a significant clinical issue. In the mouse vas deferens, a widely used model for studying sympathetic transmission, oxaliplatin triggers increased autonomic excitability, leading to bursts of neurotransmitter release from sympathetic nerves. This leads us to speculate that, in this case, oxaliplatin may have triggered adrenergic overstimulation of the heart, resulting in Takotsubo cardiomyopathy If oxaliplatin primarily targeted sympathetic fibers, this could explain the midventricular variant of TTC, instead of the typical apical form, as nerve-ending density is lower at the apex. However, the greater density of beta-adrenergic receptors at the apex may enhance the response to circulating epinephrine, potentially explaining the classic TTC shape. Still, the mechanisms behind the different patterns of TTC remain unclear [12].

Furthermore, the case presented by González-Gil et al. [13] underscores the potential for stress-induced cardiomyopathy in patients undergoing cytoreductive surgery (CRS) combined with HIPEC. It is plausible that the significant surgical stress associated with hemodynamic changes during the HIPEC phase played a major role in the development of Takotsubo cardiomyopathy. During CRS, substantial volume loss occurs due to blood loss and prolonged visceral exposure, which may contribute to hemodynamic instability. Additionally, the hyperthermic intraperitoneal chemotherapy can induce peripheral vasodilation, further exacerbating hemodynamic changes. Cytoreductive surgery combined with HIPEC also increases proinflammatory cytokine levels, promoting vasodilation and vascular permeabilization, which may interfere with hemodynamic stability. These factors, together with the stress of the extensive surgical procedure, likely contributed to the onset of Takotsubo cardiomyopathy in this patient.

In our case, the patient presented with Takotsubo cardiomyopathy after undergoing cytoreductive surgery and oxaliplatin-based HIPEC. In 2021, Keramida et al. [14] explored the association between Takotsubo cardiomyopathy and cancer in their review, emphasizing the increasing frequency of TTS among oncological patients. Their objective was to assess the incidence of TTS in cancer patients and its potential triggers, which include the malignancy itself, anticancer therapies (such as chemotherapy, immunotherapy, and radiotherapy), and emotional or physical stressors. Multiple studies highlighted a higher incidence of TTS in patients with solid tumors, especially breast and lung cancers, with rates as high as 12% in some cohorts. The review further examined the pathophysiological mechanisms of TTS, noting the involvement of sympathetic nervous system activation and coronary vasospasm. Clinical presentations often mimic acute coronary syndrome, with 26.8% of cases presenting as cardiogenic shock. The review also pointed to the negative impact of TTS on cancer treatment, as it can lead to the interruption of oncological therapies. Despite the typically benign course of TTS, its presence in cancer patients is associated with higher mortality rates and extended hospital stays. The authors called for improved risk stratification and the inclusion of TTS in the differential diagnosis of perioperative and treatment-related complications in cancer patients, highlighting the need for further research to understand its underlying mechanisms and to optimize management strategies.

In a case report by Osorio-Toro et al. [12], a 64-year-old female with stage IV metastatic gastric adenocarcinoma developed Takotsubo cardiomyopathy (TCM) during her sixth cycle of chemotherapy with oxaliplatin. The patient had no prior cardiovascular history but experienced chest pain, bradycardia, and hypotension during treatment. Cardiac tests revealed left ventricular dysfunction typical of TCM, while coronary angiography showed no significant coronary artery lesions. The case was managed with supportive care, and oxaliplatin was discontinued. The patient was successfully switched to pembrolizumab monotherapy, achieving partial response without further cardiovascular events. This case underscores the importance of monitoring for oxaliplatin-associated cardiotoxicity, particularly since oxaliplatin may act either directly or through its metabolite, oxalate, which has a chelating effect on both calcium and magnesium. This chelation could interfere with channel kinetics, potentially contributing to the onset of TCM. Therefore, it is essential to consider the possibility of oxaliplatin-induced cardiotoxicity and adopt individualized treatment strategies for managing metastatic gastric cancer in patients at risk.

In the article by Desai et al. [15], the authors review Takotsubo cardiomyopathy (TCM) in cancer patients, highlighting how stress from a cancer diagnosis, surgery, and systemic therapy may increase the risk of developing TCM. TCM is a syndrome that presents with symptoms similar to those of acute coronary syndrome or heart failure, but without coronary artery blockage. In this study, case reports of cancer patients developing TCM after receiving anti-neoplastic therapy were reviewed. Data on clinical presentation, diagnostic tests, and patient outcomes were analyzed. The review underscores that cancer patients are at heightened risk for stress-induced cardiomyopathy, largely due to physical and emotional stressors related to their diagnosis and treatment. Several chemotherapy agents, including 5-fluorouracil, capecitabine, and bevacizumab, were associated with TCM, with symptoms often occurring during the first chemotherapy cycle. Most cases showed reversible left ventricular dysfunction, though TCM could lead to severe complications like heart failure, arrhythmias, or death. This study emphasizes the importance of recognizing and managing TCM in cancer patients undergoing chemotherapy. Additionally, it discusses the clinical challenges of distinguishing between chemotherapy-induced cardiotoxicity and TCM, as both conditions can overlap.

Tini et al. [16] presented an update on cancer and Takotsubo cardiomyopathy in their 2024 review. TTS is characterized by transient left ventricular dysfunction, and its prognosis, particularly in cancer patients, is influenced by both cardiovascular (CV) and non-CV comorbidities. Cancer is a common comorbidity in TTS and is associated with increased mortality. Additionally, several anticancer therapies, including pyrimidines and immune checkpoint inhibitors (ICIs), have been linked to TTS as a form of cardiotoxicity. The review highlights the importance of collaboration between oncologists and cardiologists to manage acute TTS episodes and to safely resume cancer treatment. Despite some data supporting a link between anticancer therapies and TTS, more research is needed to understand this association fully.

In the article by Angeli et al. [17], the authors presented the complex interplay between psychosocial factors—specifically depression and anxiety—as critical cardiovascular risk determinants in women. They addressed how psychological stress exacerbates angina and is associated with adverse outcomes. They further examined sex-related discrepancies in ischemic heart disease, noting the heightened incidence of microvascular dysfunction and myocardial infarction with non-obstructive coronary arteries (MINOCA) in female patients. They concluded by advocating for a gender-specific, patient-centered diagnostic approach to enhance clinical management and address the unique challenges encountered by women in cardiovascular care.

The article “Sex Differences in Heart Failure: What Do We Know?” by Arata et al. [18] reviews the differences in heart failure (HF) between men and women, focusing on how these disparities affect diagnosis, presentation, and treatment. The authors note that, while the overall lifetime risk of developing HF is similar between sexes, women are more likely to experience heart failure with preserved ejection fraction (HFpEF), whereas men tend to develop heart failure with reduced ejection fraction (HFrEF). They highlight the role of sex-specific factors such as adverse pregnancy outcomes and premature menopause in increasing the risk of HF in women. Despite these differences, current clinical guidelines often lack sex-specific recommendations for HF management. The review emphasizes the need for sex-specific approaches in diagnosis, therapy, and clinical trials to address existing disparities in HF treatment and outcomes. The authors call for more inclusive research to fill knowledge gaps and improve the management of HF in both men and women.

## 4. Conclusions

In conclusion, this case underscores the critical need for heightened vigilance in recognizing rare but life-threatening perioperative complications [19], such as Takotsubo cardiomyopathy, in patients undergoing extensive cytoreductive surgery and HIPEC for recurrent metastatic cancer. The successful stabilization and recovery of the patient, despite the acute onset of ventricular fibrillation and cardiogenic shock, were made possible by prompt multidisciplinary intervention. Three years post-procedure, the patient exhibits no signs of oncological recurrence, highlighting the importance of integrated perioperative care in ensuring both cardiac and oncologic outcomes in high-risk cancer patients.

## Figures and Tables

**Figure 1 diagnostics-14-02402-f001:**
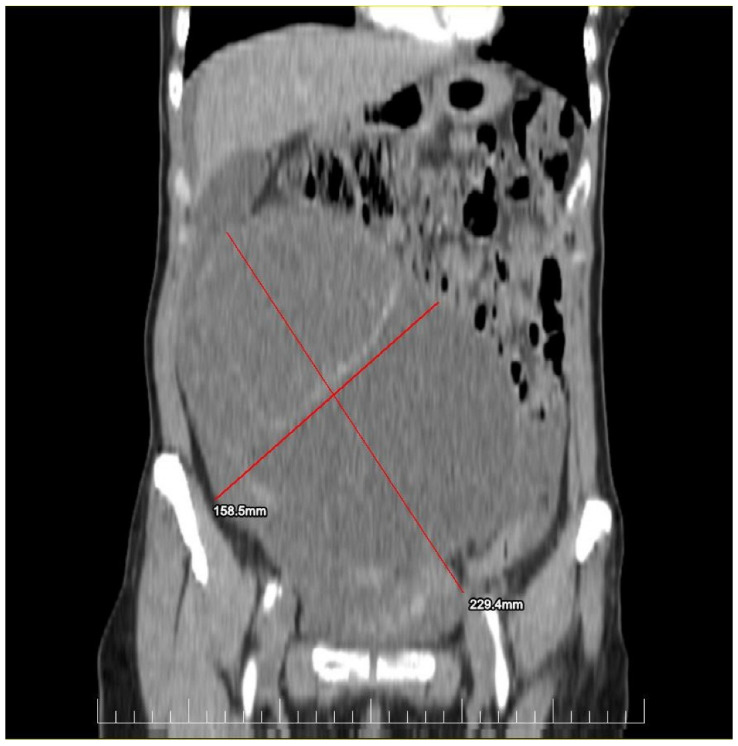
Preoperatory coronal view of a Computed Tomography (CT) image presenting a giant ovarian mass.

**Figure 2 diagnostics-14-02402-f002:**
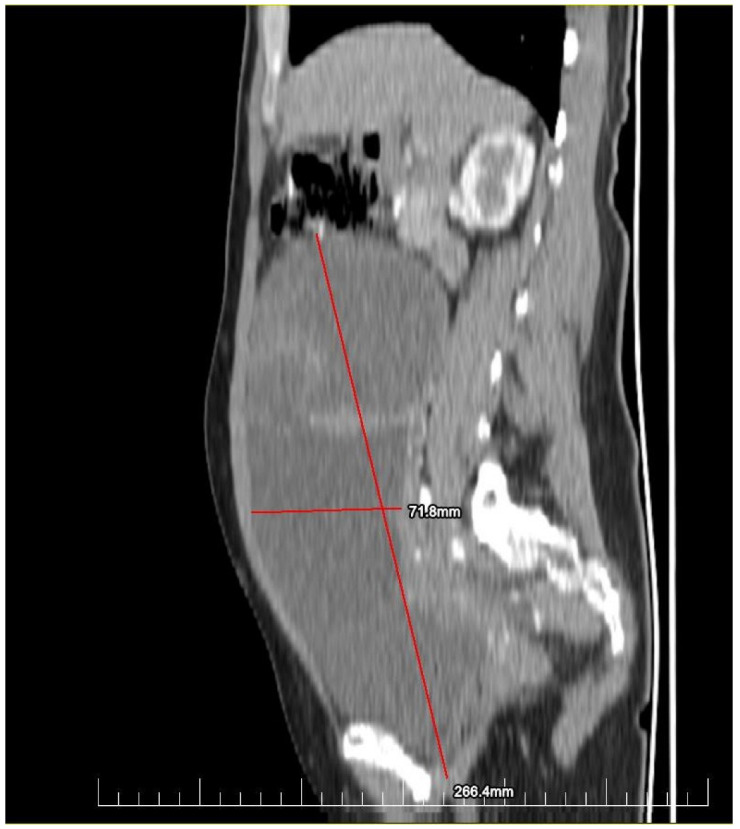
Preoperatory sagittal view of a CT image presenting a giant ovarian mass.

**Figure 3 diagnostics-14-02402-f003:**
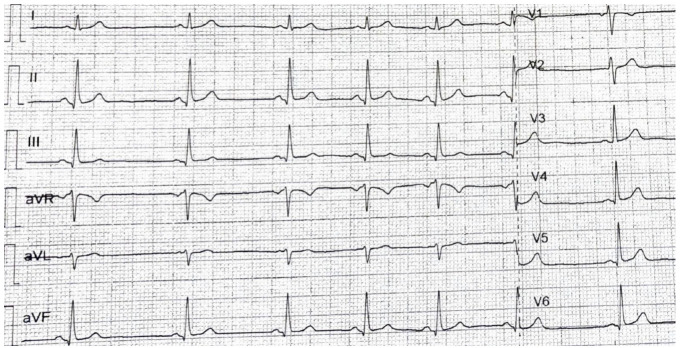
The ECG of the patient at admission, showing no signs of modifications.

**Figure 4 diagnostics-14-02402-f004:**
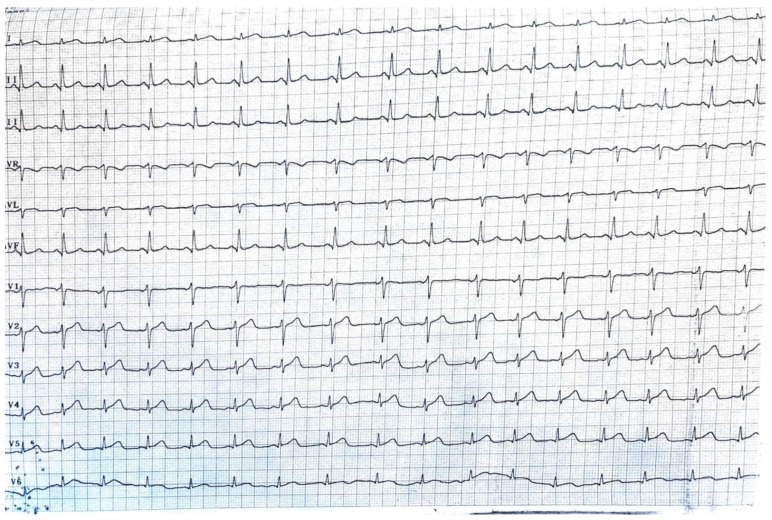
The ECG of the patient after CPR, showing no signs of modifications.

**Figure 5 diagnostics-14-02402-f005:**
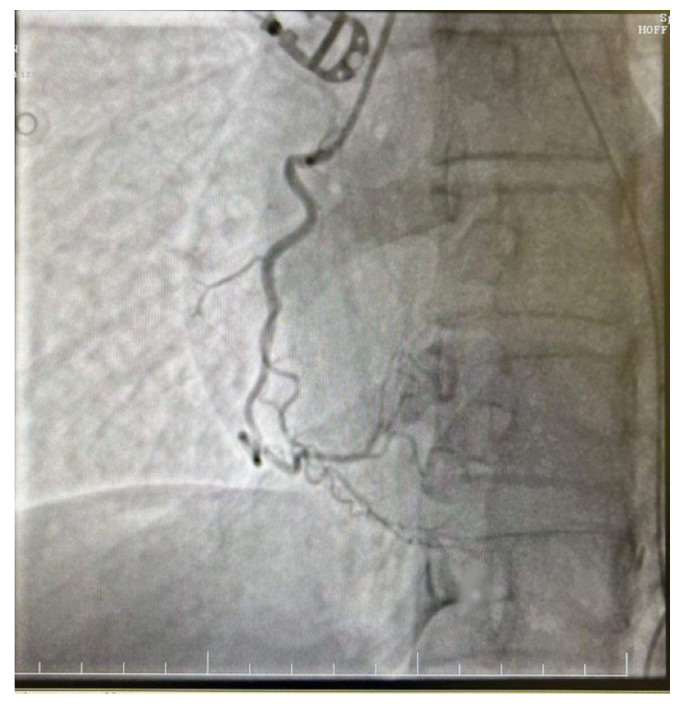
Right dominant coronary system without significant angiographic lesions.

**Figure 6 diagnostics-14-02402-f006:**
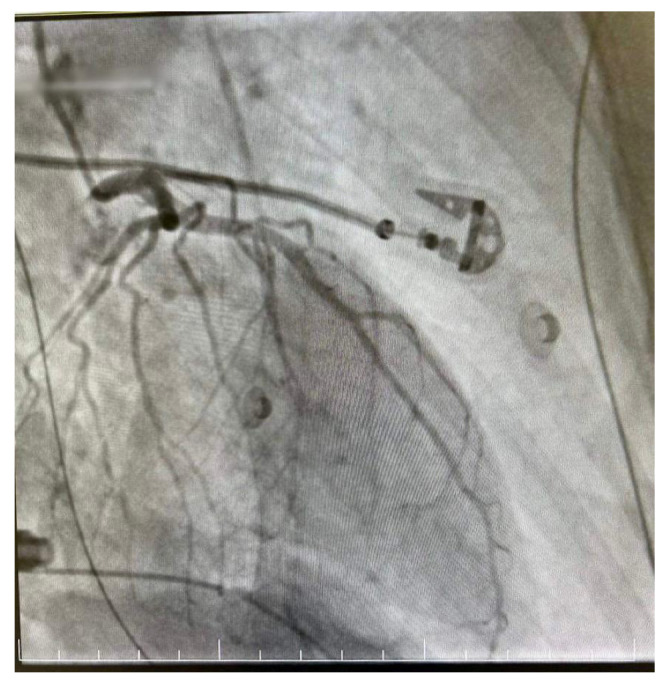
Right dominant coronary system without significant angiographic lesions.

**Figure 7 diagnostics-14-02402-f007:**
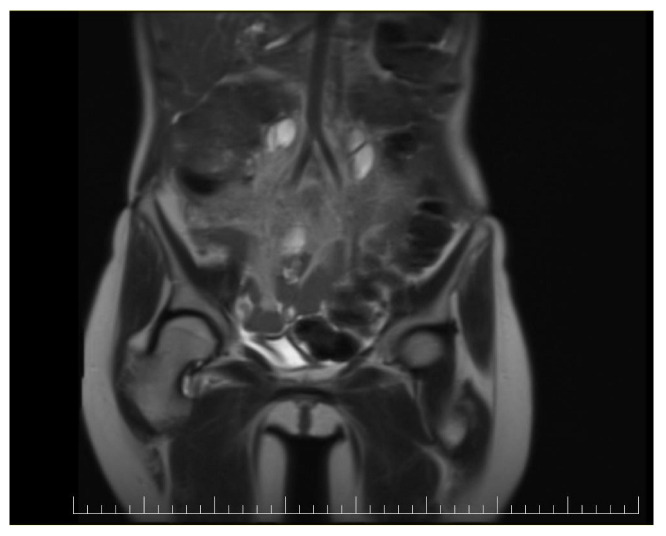
Postoperative coronal view of a CT image showing no signs of recurrence.

**Figure 8 diagnostics-14-02402-f008:**
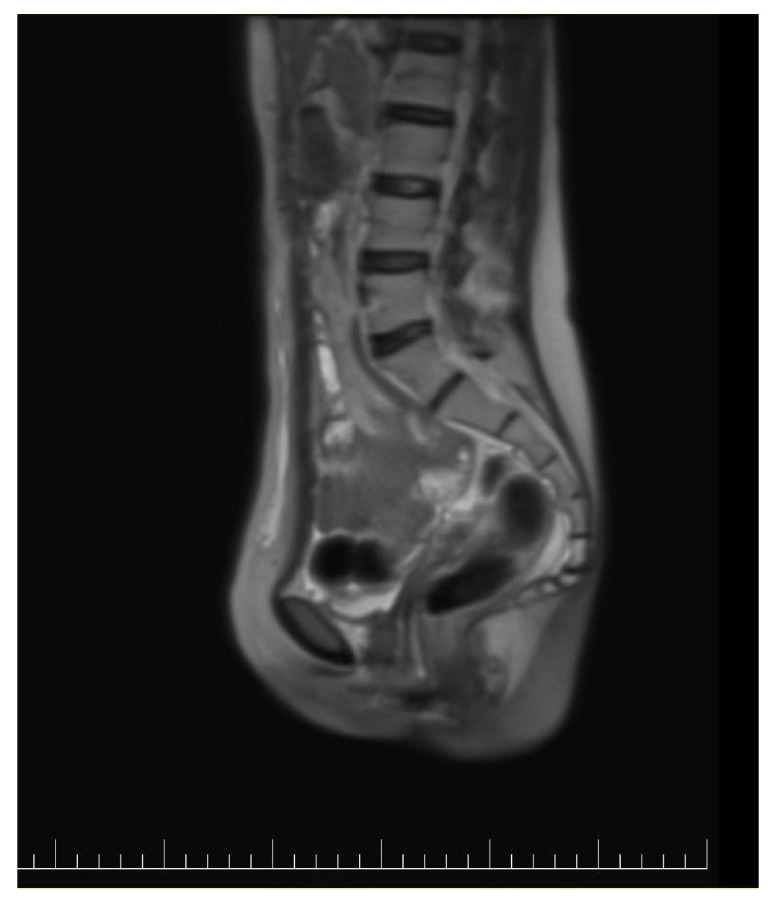
Postoperative sagittal view of a CT image showing no signs of recurrence.

## Data Availability

The original contributions presented in the study are included in the article, further inquiries can be directed to the corresponding author.

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
