# Peer review of "Takotsubo Cardiomyopathy After Cytoreductive Surgery and Hyperthermic Intraperitoneal Chemotherapy for a Recurrent Colon Cancer: A Life-Threatening Complication"

_diagnostics, 2024, doi:10.3390/diagnostics14212402_

Round 1
Reviewer 1 Report
Comments and Suggestions for Authors
Comments upon “Case reports by Moldovan B et al..
It would be more elegant if you did not use acronyms in the paper's title.
Takotsubo cardiomyopathy is now used preferably. The term syndrome is addressed more to the clinical phenotype (i.e. symptoms, signs, presentation, etc.). The term cardiomyopathy might be more appropriate as indicating that there is an inherent cardiac abnormality that predisposes to a life-threatening response to physical, emotional, or unknown stimuli. Takotsubo phenotype indicates that certain echocardiographic ventricular contractile abnormalities (primarily apical ballooning) are present.
Use of the term “broken heart syndrome” might not be appropriate, as the term “happy heart syndrome” can also be in use sometimes.
As indicated in the title, an extremely complex case is presented, while the presence of only one of the conditions might result in circulatory deterioration.
On page 2 (party background) the Authors wrote that “symptoms are mistaken for ACS…”. The sentence should be rewritten as it indicates a mistake that, actually, for clinical reasons and according to the current recommendations, requires the presence of exclusion of coronary obstruction. Thus, I propose that the Author replace the term “mistaken” with “mimicking the ACS”. One should notice that writing “mistake” is not as neutral, while “mimicking” is independent of physician notation.
The case is presented with sufficient accuracy to show the appropriateness of the diagnosis of Takotsubo cardiomyopathy and the risk of cardiac complications, as consequences of the disease and appropriate, but not free from adverse impact of the therapeutic management.
The description of the case is very comprehensive. However, I think that the demonstration of the tumor is not necessarily, a pathophysiologic explanation of the Takotsubo cardiomyopathy and does not require graphs to be present. Thus, I would advise that perioperative photos should be removed. Otherwise, one reader can focus on the tumor itself, instead of the impossible-to-vitalize processes leading to a life-threating condition.
The Authors started the discussion with the sentence “ that “HIPEC phase played a key role…”. I am not sure that the “key” word is justified. Thus I propose to write “Now, it does seem that the HIPEC did play a key role”. I would be satisfied if the Authors recall the data regarding the number of such chemotherapy in patients with recurrent colon cancer after CRS. How often cardiological complications, like those described, have been observed in these specific patients? Deeply considered and explained pathophysiological ground does not bring us to the explanation.
The case description should indicate that the current practice does not allow to be ready while facing cases like those described.
Author Response
Comment 1: It would be more elegant if you did not use acronyms in the paper's title.
Response 1: We remowed the acronyms.
Comment 2: Takotsubo cardiomyopathy is now used preferably. The term syndrome is addressed more to the clinical phenotype (i.e. symptoms, signs, presentation, etc.). The term cardiomyopathy might be more appropriate as indicating that there is an inherent cardiac abnormality that predisposes to a life-threatening response to physical, emotional, or unknown stimuli. Takotsubo phenotype indicates that certain echocardiographic ventricular contractile abnormalities (primarily apical ballooning) are present.
Response 2: We modified accordingly.
Comment 3: Use of the term “broken heart syndrome” might not be appropriate, as the term “happy heart syndrome” can also be in use sometimes.
Response 3: We modified accordingly.
Comment 4: On page 2 (party background) the Authors wrote that “symptoms are mistaken for ACS…”. The sentence should be rewritten as it indicates a mistake that, actually, for clinical reasons and according to the current recommendations, requires the presence of exclusion of coronary obstruction. Thus, I propose that the Author replace the term “mistaken” with “mimicking the ACS”. One should notice that writing “mistake” is not as neutral, while “mimicking” is independent of physician notation.
Response 4: We modified accordingly.
Comment 5: The description of the case is very comprehensive. However, I think that the demonstration of the tumor is not necessarily, a pathophysiologic explanation of the Takotsubo cardiomyopathy and does not require graphs to be present. Thus, I would advise that perioperative photos should be removed. Otherwise, one reader can focus on the tumor itself, instead of the impossible-to-vitalize processes leading to a life-threating condition.
Response 5: We remowed part of the intraoperative images, yet we would like to keep two of them presenting the ovaries, to show the big volume of the tumorous mass.
Comment 6: The Authors started the discussion with the sentence “ that “HIPEC phase played a key role…”. I am not sure that the “key” word is justified. Thus I propose to write “Now, it does seem that the HIPEC did play a key role”. I would be satisfied if the Authors recall the data regarding the number of such chemotherapy in patients with recurrent colon cancer after CRS. How often cardiological complications, like those described, have been observed in these specific patients? Deeply considered and explained pathophysiological ground does not bring us to the explanation.
Response 6: We modified accordingly. The Takotsubo Cardiomyopathy can occur both in healthy individuals, as well as in those ones which present an underlying cardiac conditions
Reviewer 2 Report
Comments and Suggestions for Authors
The authors in their paper entitled “ Takotsubo Syndrome after Cytoreductive Surgery (CRS) and Hyperthermic Intraperitoneal Chemotherapy (HIPEC) for a re current colon cancer: A life-threatening complication” provided an interesting case report regarding a case of Tako-tsubo syndrome in a female patient after oncological treatment for relapsing colon cancer.
The case is interesting and well written, however I have several observations to improve the overall quality of the manuscript:
11) Images od surgical procedures are too many; please reduce or make a single panel
22) Provide if available the ECG, echocardiography at diagnosis and coronary angiography
33) Was a ventriculography performed in the cath lab?
44) Takotsubo syndrome affects predominantly women and presents several gender related clinical differences between men and women. It should be mentioned in the discussion as it represents an important topic (please refer to these two important reviews on the topic: PMID: 38342350; PMID: 37504533)
Author Response
Comment 1: Images od surgical procedures are too many; please reduce or make a single panel
Response 1: We removed part of the images intraoperatively. We would like to keep only those ones which present the extent of the ovarian tumor.
Comment 2: Provide if available the ECG, echocardiography at diagnosis and coronary angiography
Response 2: We added images showing the ECG at admission and after CPR, also, we added images taken during angiography, bothe not presenting any signs of modifications.
Comment 3: Was a ventriculography performed in the cath lab?
Response 3: No ventriculography was performed in the cath lab, because of the altered status of the patient. The first ecocardiogram performed after the cardiac arrest, showed a modified ejection fraction of 20%, and was decided that a ventriculography would put her life in danger.
Comment 4: Takotsubo syndrome affects predominantly women and presents several gender related clinical differences between men and women. It should be mentioned in the discussion as it represents an important topic (please refer to these two important reviews on the topic: PMID: 38342350; PMID: 37504533)
Response 4: We added to the Discussion section two paragraphs which refer to the suggested articles.
Round 2
Reviewer 2 Report
Comments and Suggestions for Authors
All my observations have been answered, thank you